# Misconceptions and Knowledge Gaps on Antibiotic Use and Resistance in Four Healthcare Settings and Five European Countries—A Modified Delphi Study

**DOI:** 10.3390/antibiotics12091435

**Published:** 2023-09-11

**Authors:** Athina Chalkidou, Maarten Lambert, Gloria Cordoba, Katja Taxis, Malene Plejdrup Hansen, Lars Bjerrum

**Affiliations:** 1Section and Research Unit of General Practice, Department of Public Health, University of Copenhagen, 1353 Copenhagen, Denmark; lbjerrum@sund.ku.dk; 2Unit of PharmacoTherapy, Epidemiology and Economics, Groningen Research Institute of Pharmacy, University of Groningen, 9713 AV Groningen, The Netherlands; m.lambert@rug.nl (M.L.); k.taxis@rug.nl (K.T.); 3School of Health Sciences, University of KwaZulu-Natal, Durban 4041, South Africa; gloriac@icars-global.org; 4Center for General Practice, Department of Clinical Medicine, Aalborg University, 9220 Aalborg, Denmark; mph@dcm.aau.dk; 5Audit Project Odense, Research Unit for General Practice, Department of Public Health, University of Southern Denmark, 5000 Odense, Denmark

**Keywords:** antimicrobial resistance, antibiotics, antibiotic use, misconceptions and knowledge gaps, general practice, out-of-hours services, nursing homes, community pharmacies

## Abstract

Misconceptions and knowledge gaps about antibiotics contribute to inappropriate antibiotic use and antimicrobial resistance. This study aimed to identify and prioritize misconceptions and knowledge gaps about antibiotic use from a healthcare professionals’ perspective. A modified Delphi study with a predefined list of statements, two questionnaire rounds, and an expert meeting was conducted. The statements were rated by healthcare professionals from France, Greece, Lithuania, Poland, and Spain, and from general practice, out-of-hour services, nursing homes, and pharmacies. A total of 44 pre-defined statements covered the following themes: (1) antimicrobial resistance in general, (2) use of antibiotics in general, (3) use of antibiotics for respiratory tract infections, and (4) use of antibiotics for urinary tract infections. Consensus was defined as ≥80% agreement between the professionals during the second Delphi round. For 30% of the statements, professionals from the four settings together reached consensus. In each setting individually, at least 50% of the statements reached consensus, indicating that there are still many misconceptions and knowledge gaps that need to be addressed. Six educational tools (leaflets, posters, checklists) were developed to address the knowledge gaps and misconceptions. These can be used by patients and healthcare professionals to improve the use of antibiotics in practice.

## 1. Introduction

The rise in antimicrobial resistance (AMR) impacts the effectiveness of antibiotics dramatically [1]; a 2022 systematic analysis [2] estimated 4.95 million deaths globally in 2019 due to AMR of which about 1.27 million were directly attributable to infections caused by resistant bacteria. Inappropriate use of antibiotics, i.e., overprescribing and misuse, is the main driver of AMR [3]. To ensure the effectiveness of antibiotics in the future, it is essential to improve rational antibiotic use.

Misconceptions and lack of knowledge about antibiotics are important factors contributing to inappropriate antibiotic use [4]. According to the literature, from the patient perspective, these include false perceptions of the use and effectiveness of antibiotics and a lack of understanding of AMR. For instance, despite public campaigns, a very prevalent misconception is that antibiotics are effective against viruses, and can treat the common cold or the flu [5,6], or that they can serve as substitutes for inflammatory drugs [5]. Further, the public generally lacks knowledge about the potential side effects of antibiotics and the devastating effects of AMR [5,7].

Healthcare professionals (HCPs) often have misconceptions or a lack of knowledge about antibiotics and AMR, which may influence the way they prescribe antibiotics and their adherence to guidelines [8]. For example, a Swedish qualitative study among primary care physicians found that participants who were aware of AMR being a serious problem adhered to guidelines more strictly than those who did not consider AMR to be a big problem [8]. Further, from an HCP perspective, a perceived demand of antibiotics by the patients and fear of under-treatment are among the misconceptions that can lead to overprescribing of antibiotics [4,9]. 

To address misconceptions and knowledge gaps, it is crucial to identify these systematically and use this information to develop suitable information materials that can be implemented in antimicrobial stewardship programs. Antimicrobial stewardship programs lead to a sustainable change in the behavior of antibiotic prescribing and consumption [10] when they involve all relevant settings and local stakeholders [11,12]. Therefore, it is important to include the end-users in the design process to increase the applicability and suitability of the intervention materials [13]. 

Whilst knowledge and perceptions about antibiotic use and AMR have been thoroughly studied, such studies often focus on patients [5,7] or students [14,15] in specific settings or countries, and many assess only basic knowledge about indications and efficacy of antibiotics. There has been limited focus on specific groups of HCPs, such as prescribers [16,17] or pharmacists [18]. Furthermore, we lack insight into the knowledge gaps and misconceptions that different HCPs in the community setting in the European Union (EU) come across during their daily practice. 

This study is part of the Health Alliance for Prudent Prescribing and Yield of Antibiotics in a Patient-Centered Perspective (HAPPY PATIENT) project [19]. The HAPPY PATIENT project is an EU-funded project that aims to reduce the misuse of antibiotics in human health by developing a targeted intervention for HCPs from four healthcare settings (general practice, out-of-hours services (OoHS), nursing homes, and community pharmacies) across five target countries (France, Greece, Lithuania, Poland, Spain). For the development of the intervention and intervention material, we had to identify and prioritize misconceptions and knowledge gaps that lead to inappropriate use of antibiotics from the perspective of HCPs working in the aforementioned settings and are reported in the present study. The results of this study were used for the development of future antimicrobial stewardship interventions in the four clinical settings within the HAPPY PATIENT project. 

## 2. Results

### 2.1. Literature Search 

The literature search resulted in 46 statements representing knowledge gaps and misconceptions about antibiotic use. The statements were extracted from a total of 27 publications (Appendix A) [6,8,9,20,21,22,23,24,25,26,27,28,29,30,31,32,33,34,35,36,37,38,39,40,41,42,43]. These statements were divided into four domains: (1) AMR in general, (2) use of antibiotics in general, (3) antibiotics for acute respiratory tract infections, and (4) antibiotics for urinary tract infections. After a review of the statements by the HAPPY PATIENT project consortium members, a total of 44 final statements were included in the study.

### 2.2. Panel of Experts

Altogether, 66 experts completed the first round of the Delphi study, of whom 45 (68%) completed the second round (Table 1). Out of the 45 experts who were included in the analysis, 29 had more than 10 years of experience in their respective fields. In the general practice sector, 12 experts were general practitioners and one was a public health professional working in primary care; in the OoHS, 12 physicians participated in the panel, out of which six had a general practice background, three were pediatricians, and three worked as emergency care physicians; in the nursing home sector, four physicians completed the questionnaire, one nurse, one pharmacist working in a nursing home, and two participants did not specify; lastly, in the community pharmacy setting, twelve pharmacists participated in the study. 

### 2.3. Delphi Process

Healthcare professionals in general practice rated 39 statements to be important or very important with at least 80% consensus, which was higher than in OoHS (26 statements), nursing homes (18 statements), and pharmacies (31 statements). After the expert meeting and the second Delphi round, the number of statements that reached consensus changed in all settings. It decreased in general practice (34 statements) and increased in OoHS (30 statements), nursing homes (24 statements), and pharmacies (36 statements) (Table 2).

### 2.4. Antimicrobial Resistance

Four out of eight statements within the theme of AMR reached consensus for importance in all four settings (Table 3); for the other statements, there was no consensus in at least two settings. The experts experience that patients do not consider AMR a problem in their country and that not all antibiotics are at risk of becoming ineffective. Regarding HCPs, there is a misconception about resistance not being a problem at their own workplaces and that newly discovered antibiotics will solve the problem of resistance.

### 2.5. General Use of Antibiotics

In the four settings, consensus of importance was reached for two of nine statements (Table 3). Across all settings, there was consensus that patients believe that antibiotics are effective against all types of infections. Furthermore, there was consensus that HCPs erroneously believe that the benefits of prescribing antibiotics when unsure of the bacterial or viral origin of the symptoms outweigh the harms of exposure to antibiotics. For an additional four statements, a consensus of importance was reached in three of the four settings. 

### 2.6. The Use of Antibiotics for Respiratory Tract Infections

Of the 15 statements on the use of antibiotics for respiratory tract infections four were deemed important by the experts (Table 3). This included the only statement from the patients’ perspective that certain symptoms imply the need for antibiotics. This extends to the statements from the perspective of HCPs, where the misconceptions of sore throat with regional symptoms and fever, cough with purulent sputum, or purulent nasal discharge suggest a bacterial infection that requires antibiotics, are considered important. 

### 2.7. The Use of Antibiotics for Urinary Tract Infections

The final statements for which consensus of importance was reached in four settings were 3 of the 12 statements regarding antibiotic use for urinary tract infections (Table 3). Two of these were similar to statements for respiratory tract infections where there is a misconception about the presence of specific symptoms, which would imply the need for antibiotics. The third statement concerns the use of antibiotics to prevent complications of an uncomplicated urinary tract infection.

## 3. Discussion

### 3.1. Main Findings

In total, 44 misconceptions were identified through the literature search, 11 targeting patients and 33 targeting HCPs. Healthcare professionals in all settings recognize there are numerous misconceptions or knowledge gaps about AMR, however, the setting in which they act influences which specific misconceptions they rate as important. To a certain extent, this can be explained by the nature of the different settings; in nursing homes, sharing leftover antibiotics with family or friends is not considered a relevant misconception. Similarly, as pharmacists generally do not use diagnostic tests, diagnostic tests are not rated important in this setting. Generally, the experts reached consensus on a list of misconceptions regarding antibiotics, therewith establishing a fundament for the development of interventions to improve antibiotic use and increase knowledge.

### 3.2. Strengths and Limitations of the Study

A strength inherent to the Delphi methodology is that instead of using a random population sample, it employs a panel of experts with knowledge and experience in the field relevant to the research [44]. These experts are invited to provide their insights on a designated subject. While these insights may not always be backed up by empirical evidence, they do stem from the experts’ practical expertise [44]. Moreover, the experts are provided with feedback, enabling them to compare their individual ratings against the collective rating of the group. This process empowers them to potentially revise their opinions based on newly acquired information if they deem it necessary, and generally, it can widen the participants’ knowledge and stimulate new ideas [45]. Thus, a Delphi study generates “valid expert opinions” [44]. 

A major strength of this study is that we used input from an interdisciplinary panel of experts from five countries who have experience within their fields and in antimicrobial stewardship initiatives. Previous studies have shown that a heterogeneous sample allows for a wider range of perspectives to be taken into consideration, thus leading to better performance and increased quality and acceptance of decisions [45,46,47,48]. Additionally, the rating of the statements was conducted anonymously, thus the discussion and consensus process were not dominated by only a small group of experts [46,47,49]. 

This study has several limitations that are inherent to the Delphi methodology or introduced through modifications for the specific aims of this study. Common limitations of the Delphi methodology include the lack of a definition on how to define consensus [50], who qualifies as an expert [46,51], and the lack of agreement about the optimal number of rating rounds [44,46,47,49,50]. Additionally, the size of the Likert scale has been debated in the literature and may come with some methodological limitations [49,50,51,52]. We used a five-point Likert scale, as it ranks high in terms of reliability, validity, and ease of use. 

Lack of English fluency is a limitation of the study; however, our partners in the target countries facilitated the communication with the participants in national languages we reduced the limitation of the language barriers by translating all the statements, instructions, and all study material. 

The typical first round of “idea generation” was replaced with a literature search and the study participants received a pre-selected list of statements upon which they were invited to make a judgment. The main advantage of this modification is that the study participants start from a common base, and it allows for easier data analysis and interpretation [44]. However, this meant that the participants could not introduce new statements, which may have resulted in biased responses [44] or incomplete statements. Nevertheless, this type of bias was limited by conducting a broad and structured literature search and discussing the statements with the HAPPY PATIENT consortium. Furthermore, during the expert meetings, we provided the panel of experts the opportunity to identify any new misconceptions or knowledge gaps that might have not been taken into account during the initial round. This combination of a literature basis with practical input from the HAPPY PATIENT consortium and the participating healthcare professional is an important strength of the study, as it ensured a strong foundation for the statements. Finally, the multi-national and multi-disciplinary nature of the study meant excluding potentially relevant details for specific countries; however, this approach facilitates the creation of educational content based on the results that can be easily transferred to other settings and countries within the EU. 

### 3.3. Comparison with Existing Literature

A 2015 systematic review [53] on physicians’ knowledge, perceptions, and behavior toward antibiotic prescribing in ambulatory and hospital settings found that physicians still have inadequate knowledge about AMR and misconceptions about antibiotic prescribing. The review reports that physicians in many studies did not consider AMR to be a problem in their own clinical practice and prescribed antibiotics despite knowing that they have limited benefits for the patients. Similarly, other researchers found that HCPs are aware of AMR being a problem in general, but they do not believe it is a problem in their clinical practice [54]; HCPs prefer to overprescribe antibiotics to avoid clinical failure or when they are in doubt [9,54,55]; that national antibiotic prescription guidelines have low impact on HCPs prescribing attitudes [54,56]; and that HCPs tend to overprescribe antibiotics to meet patients’ expectations [21,24,55]. In line with other studies that were conducted in various healthcare settings, we found that some common misconceptions that need to be addressed in intervention materials are (1) that HCPs do not believe that AMR is a problem in their own practice and (2) that they often prescribe antibiotics when they are in doubt about the cause of the infection. Additionally, our study pointed out that antibiotics for urinary tract infections are overprescribed especially in nursing homes, as the HCPs often misinterpret non-specific symptoms for a urinary tract infection, or may treat asymptomatic bacteriuria more often than not, despite the fact that bacteria in the urinary tract are very common among the elderly [57].

Previous studies exploring knowledge gaps and misconceptions from the patient perspective found that it is a common misconception that antibiotics are effective against viral infections [6,36,39,41]; that they are effective against or reduce the duration of symptoms of the flu or the common cold [6,36,39,41]; that a cough of more than two weeks requires antibiotic treatment [6]; that it is fine to take antibiotics if they were used in the past to treat similar symptoms [6,41]; and that the public has limited knowledge about AMR and how it occurs [6,37,39,40,41]. Our study affirmed that all the above-mentioned patient misconceptions about antibiotics and AMR are very common issues that HCPs face in their daily practice and may lead to misuse of antibiotics.

### 3.4. Implications for Practice

Based on this study, we developed six educational tools (e.g., leaflets, posters, checklists) aimed at HCPs and the public that can be used in the different healthcare settings involved in the study (Appendix A). These tools aim to support the communication between the HCPs and the patients when they discuss their infection and the need and use of antibiotics. The tools contain information about several community-acquired infections, antimicrobial stewardship messages, the use of antibiotics, infection prevention messages, and more. Previous antimicrobial stewardship campaigns have created similar educational tools that aim to increase patient knowledge about antibiotics and AMR [58,59,60,61]. The authors encourage practicing HCPs and researchers to use the materials and to develop new materials based on this study. As our study was conducted in different countries of the EU, we believe our results and the accompanying educational tools might be useful in other parts of the EU, too. However, considering the large cultural and health structural differences between the EU and many other parts of the world, these tools might need modification for use elsewhere.

## 4. Materials and Methods

### 4.1. Design 

The study adopted a modified Delphi technique to reach consensus [17,18,19,20]. This Delphi study was modified in the form of an email survey with expert online meetings in between the two rounds, and the typical “idea generation” first round was replaced with a literature search (Figure 1). 

### 4.2. Recruitment and Coordination of Panel of Experts 

A panel consisting of experts from four different settings (general practice, OoHS, nursing homes, and community pharmacies) and five target countries (France, Greece, Lithuania, Poland, and Spain) was recruited. The local partners of the HAPPY PATIENT consortium, who are based in each target country, recruited the experts (list of project partners in Appendix A) through national and European professional associations of HCPs. The recruited experts were then asked to identify other potential experts to be included in the panel. We aimed to recruit three experts per county per setting, resulting in 60 experts to complete both rounds of the Delphi process. 

Experts were eligible and invited for inclusion if they: (i) were HCPs with knowledge and experience within the diagnosis of common community-acquired infections, antibiotic use, management of medicines, and antimicrobial stewardship programs; and (ii) had no conflicts of interest in participating in the study. There were no restrictions on English fluency, as all materials were translated into national languages. 

To overcome language barriers and ensure seamless communication between the project group and the panel of experts, the experts in each country were recruited and coordinated by the local partners associated with the HAPPY PATIENT consortium. These local partners served as the primary liaisons connecting the panel of experts with the project consortium, enabling effective communication through translation. The coordination of these local partners was overseen by A.C., responsible for coordinating the Delphi study.

### 4.3. Identification of Misconceptions and Knowledge Gaps

We conducted a literature search during May–June 2021 in PubMed, REX library system, and Google Scholar to gain an overview of the existing literature on misconceptions and knowledge gaps regarding AMR and the use of antibiotics. Search terms included “(antibiotics) AND (misconceptions)”, “(antibiotics) AND (knowledge)”, “(antimicrobial resistance) AND (misconceptions)”, “(antimicrobial resistance) AND (public knowledge)”, “(urinary tract infection) AND (misconceptions)”, “(respiratory tract infection) AND (misconceptions)”, “(respiratory tract infection) AND (diagnosis)”, “(urinary tract infection) AND (diagnosis)”. Studies were considered for inclusion if published in a peer-reviewed scientific journal in English, regardless of publication date, sample size, study design, or geographical location. A.C. and G.C. reviewed titles and abstracts to identify knowledge gaps and misconceptions that lead to inappropriate prescribing, dispensing, and consumption of antibiotics when managing community-acquired infections, from the perspective of the patient and the HCP. Succeeding the literature review, A.C. and G.C. extracted statements of potential knowledge gaps and misconceptions that may lead to inappropriate use of antibiotics. Subsequently, an iterative review process took place, where the HAPPY PATIENT project consortium reviewed the statements and suggested revisions in the phrasing and inclusion and exclusion of statements. Emphasis was placed on including statements that were relevant across all target countries.

### 4.4. Data Collection

Two rounds of data collection were implemented between September 2021 and January 2022. SurveyXact^®^ was used to distribute the online survey. A link to each survey was distributed via email to all participants by our host partners, followed by reminders in each survey round. 

#### 4.4.1. Round 1

The first round of data collection took place during September–October 2021. The experts received the pre-defined list of 44 statements (Appendix A) and they were instructed to rate the importance of each statement on a five-point Likert scale: 1 = not important, 2 = little important, 3 = neutral/I don’t know, 4 = important, 5 = very important. The experts evaluated each statement based on how important they believed the statement contributed to the misuse of antibiotics and the need to be addressed in future intervention materials. The experts were invited to prioritize common misconceptions about AMR, diagnosis, and management of community-acquired infections that HCPs face during their daily communication with patients, as well as misconceptions or knowledge gaps of their peers working in their respective healthcare settings that lead to misuse of antibiotics. Response rates were monitored regularly, and email reminders were sent to the participants who had not responded to the survey two and four weeks after the initial email invitation.

#### 4.4.2. Feedback and Expert Meetings

After the first round was completed, all participants received personalized feedback, which included the median, minimal and maximal ratings of each statement. This allowed participants to reflect on their answers based on the group median and provided the basis for the subsequent discussions during the consensus meetings.

During four meetings (one per setting) in November 2021, experts discussed disagreements, provided clarifications, and elaborated on similarities and differences across countries that would benefit the consensus process. During the meetings, no items were discarded from the survey; however, minor changes for some of the statements were suggested and implemented before the second round of the modified Delphi process. 

#### 4.4.3. Round 2 

The data collection for the second Delphi round took place during December 2021–January 2022. The second survey contained all statements of the first round. In this round, the participants had the opportunity to reassess their initial responses, based on the feedback material and the expert meeting discussions [18,21], and resubmit their rating on the same five-point Likert scale as in round 1. Response rates were monitored, and email reminders were sent to participants who had not yet completed the survey. 

### 4.5. Definition of Consensus and End of Delphi Process

Consensus was achieved if ≥80% of the participants rated a statement with 4+ on the five-point Likert scale during the second Delphi round. 

### 4.6. Data Analysis 

To identify the most important knowledge gaps/misconceptions for each of the four settings included in the study, the data analysis was conducted per setting. All data were imported and analyzed in IBM SPSS version 27. We ran descriptive statistics and calculated frequencies, median, mean and standard deviation, and minimum and maximum ratings for all the statements.

## 5. Conclusions

Experts from four different settings from five EU countries rated the most important knowledge gaps and misconceptions impacting the inappropriate use of antibiotics for the management of community-acquired infections. The results of this study give an overview of different topics that need to be addressed in educational campaigns to optimize antibiotic use. This study has resulted in the development of different intervention materials and can be used as a guide in the development of future antimicrobial stewardship programs targeting the management of community-acquired infections.

## Figures and Tables

**Figure 1 antibiotics-12-01435-f001:**
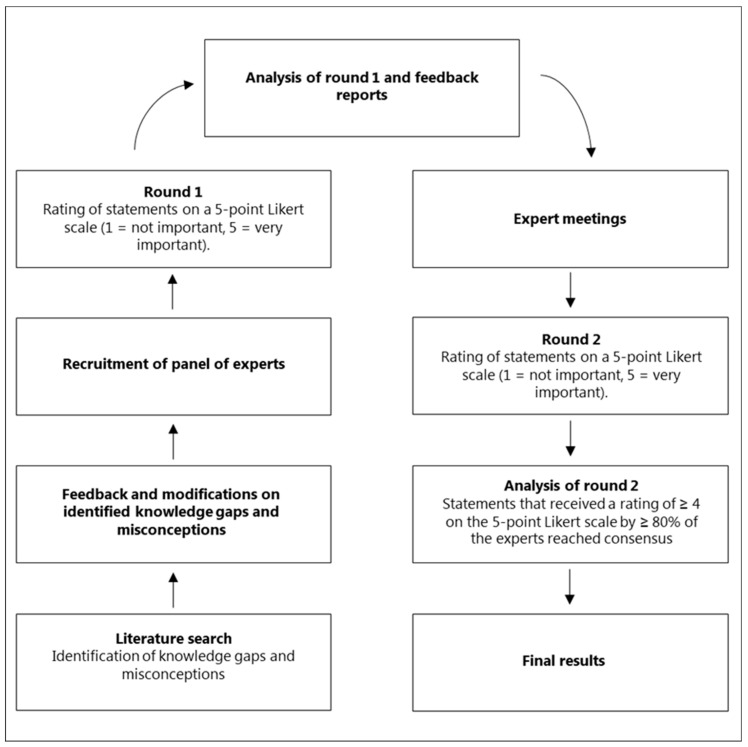
Flow chart of the study process.

**Table 1 antibiotics-12-01435-t001:** Overview of participants who completed both Delphi rounds.

Participating Countries	General Practice	Out-of-Hours Services	Nursing Homes	Pharmacies	Total
France	2	2	1	2	7
Greece	2	1	0	1	4
Lithuania	3	3	2	3	11
Poland	3	3	3	3	12
Spain	3	3	2	3	11
Total	13	12	8	12	45

**Table 2 antibiotics-12-01435-t002:** Statements that reached 80% consensus for the first and second rounds. Data are presented separately for the four settings.

	General Practice		Out-of-Hours Services		Nursing Homes		Community Pharmacies	
	Round 1 (N = 17)	Round 2 (N = 13)	Round 1 (N = 16)	Round 2 (N = 12)	Round 1 (N = 15)	Round 2 (N = 8)	Round 1 (N = 18)	Round 2 (N = 12)
Theme 1: Antimicrobial resistance (8) *	8	6	8	4	6	6	7	7
Theme 2: Antibiotic use (9) *	7	4	8	6	5	5	8	9
Theme 3: Respiratory tract infections (15) *	13	12	8	12	6	5	11	12
Theme 4: Urinary tract infections (12) *	11	12	2	8	1	8	5	8
Total (44) *:	39	34	26	30	18	24	31	36

* Total number of statements.

**Table 3 antibiotics-12-01435-t003:** Mean and percentage of consensus for all statements per setting divided by theme. Statements have reached consensus where the consensus level is ≥80%.

No	Statements Divided by Theme	Mean (Consensus Level in %)	Mean (Consensus Level in %)	Mean (Consensus Level in %)	Mean (Consensus Level in %)	4-Setting Consensus
	Theme 1: Statements related to antimicrobial resistance in general	General practice	Out-of-hours services	Nursing homes	Community pharmacies	All settings
1	Bacteria resistant to antibiotics are only present in hospitals *	4.46 (100.0)	3.92 (66.6)	4.38 (75.0)	4.46 (91.7)	
2	Antimicrobial resistance is not a problem in my country *	4.46 (84.6)	4.46 (91.7)	4.46 (87.5)	4.46 (91.7)	×
3	I cannot contribute to the increase of antimicrobial resistance *	4.38 (92.4)	4.25 (75.0)	4.38 (87.5)	4.38 (91.7)	
4	Others, not me, are responsible for controlling the problem of antimicrobial resistance †	4.23 (92.3)	4.00 (75.0)	4.23 (87.5)	4.23 (91.7)	
5	Antimicrobial Resistance is not a problem where I work †	4.15 (92.3)	4.15 (83.3)	4.15 (100.0)	4.15 (83.3)	×
6	Antimicrobial resistance is not an important problem because better antibiotics are continuously being discovered †	4.08 (84.7)	4.08 (91.7)	4.08 (100.0)	4.08 (91.7)	×
7	Not all antibiotics are at risk of becoming ineffective against infections by resistant bacteria *	4.00 (77.0)	4.00 (83.3)	4.00 (62.5)	4.00 (91.6)	×
8	If I am not exposed to antibiotics (e.g., directly by consuming antibiotics, or indirectly via the environment), then I cannot carry or transmit antibiotic-resistant bacteria *	4.00 (77.0)	4.00 (75.0)	3.85 (87.5)	3.67 (66.7)	
	Theme 2: Statements about the use of antibiotics in general	General practice	Out-of-hours services	Nursing homes	Community pharmacies	All settings
9	It is fine to use leftover antibiotics (or sharing antibiotics with family and friends) without consulting a healthcare professional, when experiencing similar symptoms to previous acute infections *	4.92 (100.0)	4.92 (91.6)	4.13 (75.0)	4.92 (100.0)	
10	The single presence of fever suggests high probability of bacterial infection and need of antibiotics *	4.46 (92.3)	4.17 (75.0)	4.46 (87.5)	4.46 (100.0)	
11	The benefits of prescribing antibiotics when unsure of the bacterial or viral origin of the symptoms outweigh the harms of exposure to antibiotics †	4.46 (84.6)	4.46 (100.0)	4.46 (100.0)	4.46 (100.0)	×
12	Antibiotics are effective against all type of infections *	4.38 (84.6)	4.38 (100.0)	4.38 (100.0)	4.38 (100.0)	×
13	Broad spectrum antibiotics, such as quinolones and 3rd–5th generation cephalosporines, are the best treatment options because they cover a wide range of bacteria †	3.85 (69.3)	3.85 (91.6)	3.85 (87.5)	3.85 (83.3)	
14	Ending the consultation without an antibiotic prescription, when the patient is asking for it, indicates lack of empathy from the doctor *	3.85 (77.0)	3.85 (91.7)	4.13 (75.0)	3.85 (91.7)	
15	Ending the consultation without an antibiotic prescription indicates that the doctor is not taking my symptoms seriously enough *	3.77 (69.3)	4.25 (91.7)	3.77 (100.0)	3.77 (91.7)	
16	Ciprofloxacin, doxycycline, levofloxacin, ofloxacin, tetracycline, trimethoprim do not cause sensitivity to sunlight †	3.23 (53.9)	3.67 (66.7)	4.00 (75.0)	3.23 (83.4)	
17	A good doctor is the one that prescribes the newest type of antibiotics †	3.15 (53.9)	3.83 (75.0)	4.13 (75.0)	3.15 (91.6)	
	Theme 3: Statements about the use of antibiotics for respiratory tract infections	General practice	Out-of-hours services	Nursing homes	Community pharmacies	All settings
18	More than 2 weeks coughing suggests a high probability of bacterial infection and need of antibiotics †	4.69 (92.3)	4.69 (100.0)	4.00 (75.0)	4.69 (100.0)	
19	As soon as I feel symptoms like sore throat, running nose, fever I should seek medical care to get antibiotics *	4.62 (92.3)	4.62 (83.3)	4.62 (100.0)	4.62 (91.6)	×
20	All children with middle ear inflammation and ear pain require antibiotic therapy †	4.54 (92.3)	4.54 (91.7)	3.50 (50.0)	4.54 (100.0)	
21	The single presence of tonsillar exudate in patients with sore throat suggests a high probability of bacterial infection and need of antibiotics †	4.46 (92.3)	4.46 (91.6)	4.00 (75.0)	4.46 (83.3)	
22	In patients with sore throat and other symptoms such as tonsillar exudates, fever, tender anterior cervical adenopathy, antibiotics have a great impact in the course of symptoms by shortening the length of symptoms by more than two days †	4.46 (100.0)	4.46 (91.7)	4.46 (100.0)	4.46 (83.4)	×
23	Based on the characteristics of the cough the health care professional can differentiate the viral or bacterial origin of the cough. For example, a chesty cough (wet, productive, or phlegmy) means that it is caused by a bacterium †	4.38 (92.3)	4.38 (91.7)	3.38 (62.5)	4.38 (83.4)	
24	A patient with the combination of two or more of the following symptoms: (a) nasal congestion, (b) nasal discharge, (c) pain in the face/teeth, (d) reduced sense of smell, (e) fever; requires antibiotic therapy independently of the number of days with symptoms †	4.31 (92.3)	4.31 (83.3)	3.63 (62.5)	4.31 (100.0)	
25	Cough with purulent sputum (or change of color of the sputum) suggests a high probability of bacterial infection and need of antibiotics †	4.31 (84.6)	4.31 (91.6)	4.31 (100.0)	4.31 (100.0)	×
26	The single presence of tender anterior cervical adenopathy in patients with sore throat suggests a high probability of bacterial infection and need of antibiotics †	4.23 (84.7)	4.00 (75.0)	4.23 (87.5)	4.17 (75.0)	
27	Purulent nasal discharge suggests a high probability of bacterial infection and need of antibiotics †	4.23 (84.6)	4.23 (91.7)	4.23 (87.5)	4.23 (100.0)	×
28	The majority of patients with a sore throat require antibiotic treatment †	4.15 (84.7)	4.15 (83.4)	3.63 (62.5)	4.15 (91.7)	
29	A bacterial infection is the most common cause of the single or combined presentation of the following symptoms: (a) nasal congestion, (b) nasal discharge, (c) pain in the face/teeth, (d) reduced sense of smell, (e) fever †	3.92 (84.6)	4.17 (75.0)	3.88 (75.0)	4.50 (83.4)	
30	The presence of cough without other symptom suggests a high probability of bacterial infection and need of antibiotics †	4.08 (77.0)	4.08 (83.3)	3.75 (62.5)	4.08 (91.7)	
31	Macrolides are the best first option for treating a bacterial lower respiratory tract infection in order to cover typical and atypical pathogens †	3.85 (69.3)	3.85 (91.6)	4.13 (75.0)	3.83 (66.7)	
32	A sinus X-ray can help doctors to discriminate the bacterial or viral origin of the rhinosinusitis symptoms †	3.31 (53.9)	3.33 (58.3)	3.50 (50.0)	3.33 (41.6)	
	Theme 4: Statements about the use of antibiotics for urinary tract infections	General practice	Out-of-hours services	Nursing homes	Community pharmacies	All settings
33	The single presence of painful discharge of urine suggests a high probability of bacterial infection and need of antibiotics †	4.69 (100.0)	4.69 (83.3)	4.69 (87.5)	4.69 (91.6)	×
34	The single presence of frequent urination suggests a high probability of bacterial infection and need of antibiotics †	4.54 (100.0)	4.54 (91.7)	4.00 (75.0)	4.54 (100.0)	
35	The single presence of burning sensation during urination suggests a high probability of bacterial infection and need of antibiotics †	4.54 (100.0)	4.54 (91.7)	3.88 (75.0)	4.54 (100.0)	
36	The single presence of blood in urine suggests a high probability of bacterial infection and need of antibiotics †	4.46 (92.3)	4.46 (83.3)	4.46 (87.5)	4.46 (100.0)	×
37	When a patient comes with acute UTI ^1^ symptoms it is okay to prescribe antibiotics, despite of the negative result of a dipstick test [nitrites (−), leucocytes (−)]. A negative dipstick test is not a good predictor of absence of UTI ^1^ †	4.46 (100.0)	4.46 (100.0)	4.46 (87.5)	3.17 (25.0)	
38	Leucocytes positive and nitrite negative result in a dipstick test indicates with high certainty bacterial infection and need of antibiotics †	4.31 (92.3)	4.00 (75.0)	4.31 (87.5)	3.58 (41.7)	
39	The single presence of smelly urine suggests a high probability of bacterial infection and need of antibiotics †	4.31 (84.7)	3.92 (75.0)	4.00 (75.0)	4.31 (83.4)	
40	The single presence of cloudy urine suggests a high probability of bacterial infection and need of antibiotics †	4.31 (84.7)	3.75 (66.7)	4.31 (87.5)	4.31 (83.4)	
41	A positive dipstick in the elderly without urinary tract symptoms is a strong indicator for urinary tract infection and requires antibiotics †	4.31 (92.3)	4.31 (83.4)	4.31 (100.0)	3.67 (50.0)	
42	The single presence of persistent urge to urinate suggests a high probability of bacterial infection and need of antibiotics †	4.23 (84.7)	4.23 (83.3)	3.75 (75.0)	4.23 (100.0)	
43	In an uncomplicated UTI ^1^, antibiotic treatment should be started as soon as possible to prevent the dissemination of the infection to the kidneys and bloodstream, independently of the risk of complication †	4.23 (84.7)	4.23 (83.3)	4.23 (87.5)	4.23 (91.7)	×
44	Cognitive changes (e.g., agitation, confusion) in the elderly suggest a high probability of bacterial infection and the need of antibiotics, even without the presence of urinary tract symptoms †	4.00 (84.7)	4.08 (75.0)	4.00 (87.5)	3.75 (58.3)	

Notes: The statements from the point of view of the healthcare professional and the patient are marked differently in the table (* From the point of view of the patient, † from the point of view of the healthcare professional). Each statement that reached consensus in all four settings is marked with an “×” in the last column of the table. ^1^ Urinary tract infection.

## Data Availability

The datasets used and/or analyzed during the current study are available from the corresponding author on request.

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
