# Peer review of "Misconceptions and Knowledge Gaps on Antibiotic Use and Resistance in Four Healthcare Settings and Five European Countries—A Modified Delphi Study"

_antibiotics, 2023, doi:10.3390/antibiotics12091435_

Round 1
Reviewer 1 Report
1. First, literature search inclusion is unnecessary for Delphi and the analysis can be based on your current results. The title needs modification and Abstract must be simple.
2. Introduction, what are the main misconception and the knowledge gaps?
2.1. How authors linked this multicountry study with the previous HAPPY PATIENT project. Are the results compared with the previous study?
2.2. Improve the introduction part as more information is needed.
3. Method, how coordination was built among the experts from different countries.
3.1. Where literature results are reported as a method for lit search is defined accurately.
3.2. Results are presented in the accepted form but need more simplification in round 2 results.
4. Add more literature from LMICs to the discussion part:
10.3389/fmed.2022.962657
Author Response
Dear reviewer,
We would like to thank you for your fast and thorough review of our manuscript. We feel that your valuable comments have surely increased the quality of our work. Please find below a point-by-point reply to all your comments. The line numbers in this document refer to the line numbers in the “tracked changes” version of the manuscript that is submitted with this document.
1. First, literature search inclusion is unnecessary for Delphi and the analysis can be based on your current results. The title needs modification and Abstract must be simple.
We agree that the inclusion of a literature search is not necessary for Delphi studies and many published Delphi studies do not go into detail about this part of the process. However, using an extensive literature search as basis for the first Delphi round is common practice1. Therefore, as we did conduct the literature search, we decided to also report on the methodology behind this to ensure more transparency and possible replicability. Further, we have revised and simplified the title and abstract.
- Jünger S, Payne SA, Brine J, Radbruch L, Brearley SG. Guidance on Conducting and REporting DElphi Studies (CREDES) in palliative care: Recommendations based on a methodological systematic review. Palliative Medicine. 2017;31(8):684-706. doi:10.1177/0269216317690685
We modified the title and the abstract as follows:
Title: Misconceptions and knowledge gaps on antibiotic use and resistance in four healthcare settings and five European countries – a modified Delphi
Abstract: Misconceptions and knowledge gaps about antibiotics contribute to inappropriate antibiotic use and antimicrobial resistance. This study aimed to identify and prioritize misconceptions and knowledge gaps about antibiotic use from a healthcare professionals’ perspective. A modified Delphi study with a predefined list of statements, two questionnaire rounds, and an expert meeting was conducted. The statements were rated by healthcare professionals from France, Greece, Lithuania, Poland, and Spain, and from general practice, out-of-hour services, nursing homes, and pharmacies. The total of 44 pre-defined statements covered the following themes: 1) antimicrobial resistance in general, 2) use of antibiotics in general, 3) use of antibiotics for respiratory tract infections, and 4) use of antibiotics for urinary tract infections. Consensus was defined as ≥80% agreement between the professionals during the second Delphi round. For 30% of the statements, professionals from the four settings together reached consensus. In each setting individually, at least 50% of the statements reached consensus, indicating that there are still many misconceptions and knowledge gaps that need to be addressed. Six educational tools (leaflets, posters, checklists) were developed to address the knowledge gaps and misconceptions. These can be used by patients and healthcare professionals to improve the use of antibiotics in practice.
2. Introduction, what are the main misconception and the knowledge gaps?
We have added some more information in the introduction part regarding misconceptions and knowledge gaps of antibiotics as we agree a more extensive introduction was appropriate (page 2 lines 51-66).
“Misconceptions and lack of knowledge about antibiotics are important factors contributing to inappropriate antibiotic use [4]. According to the literature, from the patient perspective, these include false perceptions of use and effectiveness of antibiotics and lack of understanding of AMR. For instance, despite public campaigns, a very prevalent misconception is that antibiotics are effective against viruses, and they can treat the common cold or the flu [5,6], or that they can serve as substitutes for inflammatory drugs [5]. Further, the public generally lacks knowledge about the potential side effects of antibiotics and the devastating effects of AMR [5,7].
Healthcare professionals (HCPs) often have misconceptions or a lack of knowledge about antibiotics and AMR, that may influence the way they prescribe antibiotics and their adherence to guidelines [8]. For example, a Swedish qualitative study among primary care physicians found that participants who were aware of AMR being a serious problem adhered to guidelines more strictly than those who did not consider AMR to be a big problem [8]. Further, from a HCP perspective, a perceived demand of antibiotics by the patients and fear of under-treatment are among the misconceptions that can lead to overprescribing of antibiotics [4,9].”
We also discuss this topic more extensively in the discussion section (page 9, lines 224-255).
“A 2015 systematic review [53] on physicians’ knowledge, perceptions, and behavior towards antibiotic prescribing in ambulatory and hospital settings found that physicians still have inadequate knowledge about AMR and misconceptions about antibiotic pre-scribing. The review reports that physicians in many studies did not consider AMR to be a problem in their own clinical practice and prescribed antibiotics despite knowing that they have limited benefit for the patients. Similarly, other researchers found that HCPs are aware of AMR being a problem in general, but they do not believe it is a problem in their clinical practice [54]; that HCPs prefer to overprescribe antibiotics to avoid clinical failure or when they are in doubt [9,54,55]; that national antibiotic prescription guidelines have low impact on HCPs prescribing attitudes [54,56]; and that HCPs tend to overprescribe antibiotics to meet patients’ expectations [21,24,55].
In line with other studies that were conducted in various healthcare settings, we found that some common misconceptions that need to be addressed in intervention mate-rials are 1) that HCPs do not believe that AMR is a problem in their own practice and 2) that they often prescribe antibiotics when they are in doubt about the cause of the infection. Additionally, our study pointed out that antibiotics for UTIs are overprescribed especially in the nursing homes, as the HCPs often misinterpret non-specific symptoms for UTI, or may treat asymptomatic bacteriuria more often than not, despite the fact that bacteria in the urinary tract are very common among the elderly [57].
Previous studies exploring knowledge gaps and misconceptions from the patient perspective found that it is a common misconception that antibiotics are effective against viral infections [6,36,39,41]; that they are effective against or reduce the duration of symptoms of the flu or the common cold [6,36,39,41]; that a cough of more than two weeks requires antibiotic treatment [6]; that it is fine to take antibiotics if they were used in the past to treat similar symptoms [6,41]; and that the public has limited knowledge about AMR and how it occurs [6,37,39–41]. Our study affirmed that all the above-mentioned patient misconceptions about antibiotics and AMR are a very common issue that HCPs face in their daily practice and may lead to misuse of antibiotics.”
2.1. How authors linked this multicountry study with the previous HAPPY PATIENT project. Are the results compared with the previous study?
The authors of the present study are part of the EU funded project called HAPPY PATIENT. The HAPPY PATIENT project aims to decrease the use of antibiotics in five European countries and across four different settings. In order to develop an intervention for the project, we had to understand the context specific details, including the most important misconceptions and knowledge gaps about antimicrobial resistance and the use of antibiotics across the different countries and in the found healthcare sectors involved. Therefore, the present manuscript reports on the results of the Delphi study that we conducted between autumn 2021 and spring 2022. The results of our study were used for the development of the intervention material for the HAPPY PATIENT project. Thus, the HAPPY PATIENT project was not a previous study, it’s the project that this Delphi study is part of. For more details, you can read the published protocol:
Bjerrum, A.; García-Sangenís, A.; Modena, D.; Córdoba, G.; Bjerrum, L.; Chalkidou, A.; Lykkegaard, J.; Hansen, M.P.; Søndergaard, J.; Nexøe, J.; et al. Health Alliance for Prudent Prescribing and Yield of Antibiotics in a Patient-Centred Perspective (HAPPY PATIENT): A before-and-after Intervention and Implementation Study Protocol. BMC Primary Care 2022, 23, 1–11, doi:10.1186/S12875-022-01710-1/TABLES/3.
We have revised the section in the manuscript to ensure that this information is clear. You can find the changes on page 2 lines 81-94 and below:
“This study is part of the Health Alliance for Prudent Prescribing and Yield of Antibiotics in a Patient-Centered Perspective (HAPPY PATIENT) project [16]. The HAPPY PATIENT project is a European Union funded project that aims to reduce the misuse of antibiotics in human health by developing a targeted intervention for HCPs from four healthcare settings (general practice, out-of-hours services (OoHS), nursing homes, and community pharmacies) across five target countries (France, Greece, Lithuania, Poland, Spain). For the development of the intervention and intervention material we had to identify and prioritize misconceptions and knowledge gaps that lead to inappropriate use of antibiotics from the perspective of HCPs working in the aforementioned settings and are reported in the present study. The results of this study will be used for the development of antimicrobial stewardship interventions in the four clinical settings within the HAPPY PATIENT project.”
2.2. Improve the introduction part as more information is needed.
We have added more information in the introduction part as per the above comments (2 and 2.1).
3. Method, how coordination was built among the experts from different countries.
We have revised the relevant section in the manuscript and added the following paragraph that better explains how coordination was built among the experts from different countries.
Please refer to page 12, lines 296-302:
“To overcome language barriers and ensure seamless communication between the project group and the panel of experts, the experts in each country were recruited by the local partners associated with the HAPPY PATIENT consortium. These local partners served as the primary liaisons connecting the panel of experts with the project consortium, enabling effective communication through translation. The coordination of these local partners was overseen by A.C., responsible for coordinating the Delphi study.”
3.1. Where literature results are reported as a method for lit search is defined accurately.
The results of the literature search are reported under “2.1 Literature search”. The publications used to create the statements are provided in a supplementary file. The main result of the literature search are the statements, all statements that have been created are provided in Table 3 together with the rating of the experts. We decided not to report the list of statements separately under “2.1 Literature search” to minimize duplication throughout the manuscript. We have made the following revisions in the methods section and discussion to clarify our literature search approach and its meaning for the study, that you can find on page 9 lines 205-218, and page 12 lines 305-315. You can also see our revisions below:
Methods, page 12 lines 305-315
In the methods we have changed the wording from “literature review” to “literature search” as it describes more accurately our approach.
“We conducted a literature search during May-June 2021 in PubMed, REX library system, and Google Scholar to gain an overview of the existing literature on misconceptions and knowledge gaps regarding AMR and the use of antibiotics. Search terms included “(antibiotics) AND (misconceptions)”, “(antibiotics) AND (knowledge)”, “(antimicrobial resistance) AND (misconceptions)”, “(antimicrobial resistance) AND (public knowledge)”, “(urinary tract infection) AND (misconceptions)”, “(respiratory tract infection) AND (misconceptions)”, “(respiratory tract infection) AND (diagnosis)”, “(urinary tract infection) AND (diagnosis)”. Studies were considered for inclusion if published in a peer-reviewed scientific journal in English, regardless of publication date, sample size, study design or geographical location….”
Discussion, page 9 lines 205-218
The typical first round of “idea generation” was replaced with a literature search and the study participants received a pre-selected list of statements upon which they were invited to make a judgement. The main advantage of this modification is that the study participants start from a common base, and it allows for an easier data analysis and interpretation [44]. However, this meant that the participants could not introduce new statements, which may have resulted in biased or incomplete statements [44]. Nevertheless, this type of bias was limited by conducting a broad and structured literature search and discussing the statements with the HAPPY PATIENT consortium. Furthermore, during the expert meetings we provided the panel of experts the opportunity to identify any new misconceptions or knowledge gaps that might have not been taken into account during the initial round. This combination of a literature basis with practical input from the HAPPY PATIENT consortium and the participating healthcare professionals is an important strength of the study, as it ensured a strong foundation of the statements.
3.2. Results are presented in the accepted form but need more simplification in round 2 results.
Thank you very much for pointing this out. Indeed, there was a mistake with the formatting of Table 3. It has now been revised and simplified.
Please, refer to Table 3 on page 4.
4. Add more literature from LMICs to the discussion part:
It is important to address the impact of studies on LMICs, definitely on the topic of antibiotic use and antimicrobial resistance as in our study. We would like to point out that the literature we referred to includes worldwide systematic reviews, covering countries, among others, such as Bangladesh, China, DR Congo, India, Peru, Sudan and Trinidad and Tobago. As this study focused on five countries in the European setting, we believe that our results may be extrapolated to other countries within the EU and that our educational materials might also be useful there. However, we have noticed in our study that even within the EU there are large differences in healthcare practice and culture that affect the opinions of experts. Such differences would be even greater between the EU and other places around the world, whether those are LMICs or not. Even more so, we believe that considering LMICs as one group of countries with similar cultures and healthcare systems would be very far from true. Hence, we have decided to stay close to the setting we have performed our research in as to the implications of our research. We have clarified this more under “3.4 implications for practice”.
You can find our modifications below and on page 10, lines 257-271.
“Based on this study, we developed six educational tools (e.g. leaflets, posters, checklists) aimed at HCPs and the public that can be used to improve antibiotic use (Supplementary file S2). These tools aim to support the communication between the HCPs and the patients when they discuss their infection and the need and use of antibiotics. The tools contain information about several community-acquired infections, antimicrobial stewardship messages, use of antibiotics, infection prevention messages, and more. Previous antimicrobial stewardship campaigns have created similar educational tools that aim to increase patient knowledge about antibiotics and AMR [58–61]. The authors encourage practicing HCPs and researchers to use the materials and to develop new material based on this study. As our study was conducted in different countries of the EU, we believe our results and the accompanying educational tools might be useful in other parts of the EU too. However, considering large cultural and health structural differences between the EU and many other parts of the world, these tools might need modification for use elsewhere.”
Reviewer 2 Report
Congratulations the authors for the excellent work! The antibiotic resistance is a very actual issue and the education of healthcare professional is a crucial point, so this project points out what is about the work that has yet to be done for the awareness on this problem! Well done! Well written and well performed! Very very good!
I suggest only some typos:
- Line 73 Typo 1: This supplementary file appeared for the first time so that please rename it S1. As a consequence, rename all supplementary file in the order of presentation in the manuscript. Typo 2: please converte square bracket in round bracket.
- Please revised the Table 3 for all the four themes as regards the rows 2,4,6 and 8 that are disallineate as respect to the corresponding values.
Author Response
Dear reviewer,
We would like to thank you for your very positive comments and for recognizing the importance of our study.
Also, thank you for kindly pointing out some typos in the manuscript.
- The supplementary file names have been revised, both within the text and their names. Kindly, refer to lines 99, 260, 287, 331.
- We have revised and simplified Table 3 (page 4), as indeed there was a mistake with the formatting. Thank you very much for your kind comment. We hope that the revised version is more readable and comprehensible.
Reviewer 3 Report
The authors provided information to use in educational campaigns for patients and healthcare staffs to improve the use of antibiotics. This work is significant in the clinical settings. However, expressions are unclear and the intension of authors is difficult to understand. Describe more concretely in the section of Abstract and Discussion. Moreover, the authors mentioned that the strength of this study was the use input from a panel of experts. The extracted input is not scientific. Describe scientific reasons. There are others in the following;
1. Abstract, Consensus was reached for more than half of the statements within each setting: What was it meant?
2. Method: Describe the experts in detail. What careers, howe many, etc.
Author Response
Dear reviewer,
We would like to thank you for your fast and thorough review of our manuscript. We feel that your valuable comments have surely increased the quality of our work. Also, thank you for your kind comments and for recognizing that the work we have done with this study is significant. Please find below a point-by-point reply to all your comments. The line numbers in this document refer to the line numbers in the “tracked changes” version of the manuscript that is submitted with this document.
“However, expressions are unclear and the intension of authors is difficult to understand. Describe more concretely in the section of Abstract and Discussion.”
We have made major changes in the abstract and we hope this made it clearer and easier to understand. Further, we have created a sub-section in the Discussion with the title “3.4. Implications for practice” (page 13, lines 257-271) and moved the relevant text (educational material) under this section, to separate it from the rest of the discussion. We hope this change increased comprehension of the Discussion section.
Moreover, the authors mentioned that the strength of this study was the use input from a panel of experts. The extracted input is not scientific. Describe scientific reasons.
Indeed, we consider it a major strength of this study that we involved a multi-cultural and multi-disciplinary panel of experts coming from the five countries that are participating in the HAPPY PATIENT project (the project that our study is part of). Further, to our knowledge, there hasn’t been a similar Delphi study involving such a broad panel of experts within the EU, and reporting results both per setting and across all settings, and we hope that this knowledge will strengthen antibiotic stewardship initiatives within the EU.
We have used the Delphi methodology as it is a well-known and extensively reported method in the scientific literature for a large variety of topics. As we have sticked closely to the guidelines of conducting and reporting Delphi studies1, we believe that the data extracted from the Delphi questionnaires would be considered scientific. The strength we refer to is not that we used input from a panel of experts, as this is inherent to any Delphi study. The strength of our study is the unique composition of our panel of experts, from five different EU countries and from four different healthcare settings. We agree with you that this may not have been clearly mentioned in the manuscript, so we have revised accordingly:
Page 9, lines 173-183
“A strength inherent to the Delphi methodology is that instead of using a random population sample, it employs a panel of experts with knowledge and experience in the field relevant to the research [44]. These experts are invited to provide their insights on a designated subject. While these insights may not always be backed up by empirical evidence, they do stem from the experts’ practical expertise [44]. Moreover, the experts are provided with feedback, enabling them to compare their individual ratings against the collective rating of the group. This process empowers them to potentially revise their opinions based on newly acquired information if they deem it necessary, and generally it can widen the participants’ knowledge and stimulate new ideas [45]. Thus, a Delphi study generates “valid expert opinions” [44].”
- Keeney, S.; Hasson, F.; Mckenna, H. The Delphi Technique in Nursing and Health Research. The Delphi Technique in Nursing and Health Research 2010, doi:10.1002/9781444392029.
1. Abstract, Consensus was reached for more than half of the statements within each setting: What was it meant?
Based on one of your previous suggestions and that of another reviewer, we have made major changes to the abstract. Please see below.
Abstract: Misconceptions and knowledge gaps about antibiotics contribute to inappropriate antibiotic use and antimicrobial resistance. This study aimed to identify and prioritize misconceptions and knowledge gaps about antibiotic use from a healthcare professionals’ perspective. A modified Delphi study with a predefined list of statements, two questionnaire rounds, and an expert meeting was conducted. The statements were rated by healthcare professionals from France, Greece, Lithuania, Poland, and Spain, and from general practice, out-of-hour services, nursing homes, and pharmacies. The total of 44 pre-defined statements covered the following themes: 1) antimicrobial resistance in general, 2) use of antibiotics in general, 3) use of antibiotics for respiratory tract infections, and 4) use of antibiotics for urinary tract infections. Consensus was defined as ≥80% agreement between the professionals during the second Delphi round. For 30% of the statements, professionals from the four settings together reached consensus. In each setting individually, at least 50% of the statements reached consensus, indicating that there are still many misconceptions and knowledge gaps that need to be addressed. Six educational tools (leaflets, posters, checklists) were developed to address the knowledge gaps and misconceptions. These can be used by patients and healthcare professionals to improve the use of antibiotics in practice.
2. Method: Describe the experts in detail. What careers, howe many, etc.
Indeed, it is important to provide a more detailed description of the experts participating in the Delphi study. We have revised the relevant section in the text (page 3, lines 104-115) as follows, and we hope that this revision is acceptable for you:
“Altogether, 66 experts completed the first round of the Delphi study, of whom 45 (68%) completed the second round (Table 1). Out of the 45 experts who were included in the analysis, 29 had more than 10 years of experience in their respective field. In the General Practice sector, twelve experts were general practitioners and one was a public health professional working in primary care; in the OoHS twelve physicians participated in the panel, out of which six had a general practice background, three were pediatricians, and three worked as emergency care physicians; in the nursing home sector, four physicians completed the questionnaire, one nurse, one pharmacist working in a nursing home, and two participants did not specify; lastly, in the community pharmacy setting, twelve pharmacists participated in the study.”
An overview of the number of participants broken down by country and healthcare setting can be found in Table 1 (page 3).
Round 2
Reviewer 3 Report
The authors revised appropriately. No further correction is necessary.